



# The Role of Bioenergy and Carbon Capture and Storage (BECCS) in the Case of Delayed Climate Policy – Insights from Cost-Risk Analysis

Jana Mintenig, Mohammad M. Khabbazan, and Hermann Held

Research Unit Sustainability and Global Change, Center for Earth System Research and Sustainability, Universität Hamburg, Grindelberg 5, 20144 Hamburg, Germany.

*Correspondence to:* Mohammad M. Khabbazan (mohammad.khabbazan@uni-hamburg.de)

**Abstract.** Cost-Risk Analysis (CRA), a hybrid of Cost-Effectiveness Analysis (CEA) and Cost-Benefit Analysis (CBA), has been proposed as an alternative to CEA as a decision criterion for evaluating climate policy. It weighs mitigation costs against associated risks of violating a predefined temperature guardrail, thereby enabling an analysis of otherwise infeasible temperature targets. Under CEA, delaying climate policy causes infeasibility of temperature targets which was resolved by the assessment under CRA. Indeed, CRA enables a quantitative evaluation of any delay scenario, thereby yielding information of the severeness of postponing climate policy. Alternatively, negative emission technologies have been included in CEA to enlarge the leeway in decision making and postpone infeasibility. This study closes the loop by evaluating the impact of the technology option BECCS (Bioenergy and Carbon Capture and Storage) in light of delayed climate policy under CRA. The work is conducted using the Integrated Assessment Model MIND (Model of Investment and Technological Development). This interplay creates the following insights: An inclusion of BECCS avoids corner solutions that were previously identified for delay scenarios, yielding a larger window of opportunity for action to mitigate climate change. Moreover, it postpones mitigation efforts into the future and removes the pressure to shut down fossil fuel use immediately. Thereby, mitigation-induced welfare losses are reduced substantially. BECSS, when evaluated under CRA, has confirmed well-known results from CEA. However, in contrast to results derived from CEA, mitigation-induced welfare losses decline with delay, while climate risk-induced welfare losses increase with delay by approximately the same magnitude. Hence within CRA, BECCS reduces the welfare effect of delayed climate policy by an order of magnitude. This underlines the crucial role of BECCS for the case of delay, even if one changes the decision-analytic framework from CEA to CRA and thereby softened the temperature target.

**Keywords:** BECCS, Climate Targets, Cost-Risk Analysis, Decision under Uncertainty

## 1 Introduction

From the point of view of climate policy, the 21$^{\text{st}}$ Conference of the Parties (COP) in Paris was a success because it managed to reconcile the interests of all member states, culminating in the signature of a single agreement. This agreement is structured bottom-up relying on voluntary contributions combined with the requirement to not reduce already promised efforts. However, most countries so far do not comply with their already adopted pledges. Instead of the hoped for emissions reduction the IPCC



assessment report (2014) even identifies a continuous increase of Greenhouse Gas (GHG) emissions in the past years. The most recent country specific pledges aimed at reaching the 2°C target are insufficient and may lead to a warming of around 2.7°C (IEA, 2015). Hence, the question remains whether the determined target is reachable, in the absence of other technology options such as climate engineering or BECCS, despite the celebrated Paris agreement.

The motivation for limiting global mean temperature to 2°C compared to pre-industrial levels was based upon a scientific background in combination with the precautionary principle (UNFCCC, 2011). For the climate problem two types of decision criteria are used most often. The first type is called Cost-Benefit Analysis (CBA) which weighs costs of climate change mitigation against the benefits of avoided climate damages. The second type is called Cost-Effectiveness Analysis (CEA) which employs a climate target as a strict constraint. The argument in favour of CEA is that the spectrum of potential climate damages

is unknown. Therefore, a target such as 2°C warming is defined as a first step that supposedly avoids dangerous consequences for the climate system. It reflects a precautionary approach, representing an alternative to CBA to be used as long as the option of formally defining damage functions remains rather complicated. In a second step, different paths to reach 2°C can be identified and compared based on their cost-effectiveness. In this work, the argument for CEA is supported. Nevertheless, we recognize that CEA can be problematic when uncertainty such as that in estimates of climate sensitivity comes into play.

Decision makers might encounter situations with negative expected value of information or infeasible solutions (Schmidt et al., 2011). Infeasibility would require acknowledgement of failure of the chosen approach and is particularly delicate because it would reduce the credibility of temperature targets as such.

From a political point of view, therefore, an alternative called Cost-Risk Analysis (CRA) was suggested that circumvents the inconsistencies of CEA (Schmidt et al., 2011 and Neubersch et al., 2014). Instead of including climate damage functions,

CRA weighs the risks of transgressing a predefined temperature target against the costs of mitigation, representing a hybrid of CBA and CEA. Thereby, an evaluation of the 2°C target remains possible, even in cases that would be infeasible under CEA (Neubersch et al., 2014). The value system of the 2°C target can be extrapolated into any future scenario.

Infeasibility is a major issue in the case of delayed climate policy. With CRA one can still identify the actions a decision maker might take when confronted with unviable targets. Additionally, it is possible to quantify the effects of such delay

scenarios. A major issue they encountered by Roth et al. (2015), during such an analysis, was that in most cases only corner solutions could be found. Temperature targets were transgressed for most delay scenarios and the determined policy was to seek to minimize them with the maximum allowable mitigation effort, while still adhering to the other model restrictions.

Negative emission technologies such as direct air capture in combination with carbon storage or bioenergy usage in combination with carbon capture and storage (BECCS) are options to provide the decision maker with greater flexibility in solving

the climate problem. They allow for deliberately reducing the level of carbon in the atmosphere through human intervention. Negative emissions have been included in previous research in order to solve the issue of infeasibility under CEA (Clarke et al., 2009; Riahi et al., 2013; Luderer et al., 2013; Clarke et al., 2014). But the modelled decision maker under CEA in otherwise infeasible models lacks the flexibility to choose whether to implement carbon dioxide removal technologies (CDR) or transgress the temperature target. Under CRA, however, the decision maker is free to decide to what extent to transgress the target or

to invest into mitigation. Transgression is penalised in a manner consistent with the 2°C target. The following work therefore





fills this gap in the literature by evaluating the effect of negative emissions, by the example of BECCS, under the decision framework CRA. The research question therefore asks is a decision maker would decide to rather transgress the climate target or invest into negative emissions to avoid these, given that they support the 2°C target? Furthermore, the analysis provides a quantification of the efforts.

The technology option BECCS is chosen as a representative CDR method. It involves coupling two different technologies. The first component, bioenergy, extends the portfolio energy production options. If the assumption of carbon neutrality is satisfied, it allows for carbon free energy production. The carbon neutrality assumption requires that the carbon removed during the growth phase of biomass offsets the amount of carbon released during combustion. The second component CCS (Carbon Capture and Storage) captures the emissions during the combustion process, instead of releasing them into the atmosphere,

followed by underground storage. Hence, if these two components are used simultaneously, the associated emission amount is ultimately negative. Thereby, the overall level of already accumulated emissions can be reduced. Hence, negative emission technologies open a variety of future development paths: For the 2°C target, BECCS increases the probability of reaching the target and can, thus, become an option to buy time and enable a smoother transition from fossil fuels to renewables at reduced costs. Alternatively, BECCS can be used to pursue even more stringent targets. This is in line with the desire to limit global

warming to well below 2°C or even "pursuing efforts to limit the temperature increase to 1.5°C" (COP 21, 2015, p. 2)[1].

Regardless of which temperature target is chosen, in reality BECCS can only be presented as a valid option, if significant investments into research and development are undertaken. Currently the use of both, bioenergy and CCS, involves uncertainties. For bioenergy the question arises whether growth can be managed sustainably. One of the main uncertainties regarding CCS is the leakage rate influenced by geographical properties of the storage sites and the time lag before effects can be identified and

mitigated. These technical constraints are dealt with by including hard constraints on the respective aspects. The analysis in this paper is done in the context of the decision making of a perfectly rational social planner, therefore consequences such as technological lock-in or political inertia are not under consideration in this work. A critical comment on such risks associated with BECCS can be found in Fuss et al. (2014).

Before moving towards more stringent targets, we first study the impact of BECCS under already established and accepted

targets, in order to be able to compare the impacts that the inclusion of BECCS has on the prescribed climate policy. The purpose of this study is to extend the literature on CRA (Schmidt et al., 2011 and Neubersch et al., 2014) and especially in the context of delayed climate policy (Roth et al., 2015) by additionally including BECCS. Thus, the main focus is on deriving the effect of BECCS in CRA with delayed climate policy.

The remainder of this work is structured as follows: Sect. 2 provides a short literature review on the technology option

BECCS followed by a description of the model used in this paper and a short introduction on CRA and delayed climate policy in Sect. 3. Section 4 deals with the application and a sensitivity study on bioenergy potential. Finally Sect. 5 concludes the paper.

---

[1]Just recently, the IPCC has decided to release a special report on a 1.5°C target in 2018 (IPCC, 2016).



## 2 Literature review

BECCS is composed of the two technologies, bioenergy (BE) and Carbon Capture and Storage (CCS), that can also be used independently of each other. Bioenergy is another form of renewable energy production that refers to energy generated through combustion of biomass. Biomass is characterized as "any non-fossilized material of biological origin" (Gough and Upham, 2010, p. 6). It is distinguished from other forms of renewable energy by involving carbon and the possibility of being stored and traded globally. It can be sourced as a waste product, harvested directly from natural ecosystems or a crop designated for energy usage. Several components need to be considered for the evaluation of bioenergy. Its main uncertainties relate to resource potentials and deployment. Assumptions have to be made with regard to a wide range of factors such as "soil degradation; water scarcity; yield growth; production potential of degraded land; nature protection; and climate change feedback" (Creutzig et al., 2012b, p. 70). Hence, the main question is to what respect sustainability considerations are taken into account.

CCS is a technology that captures carbon during the combustion process, compresses it and in a last step stores it in deposits[2] underground. It is not a standalone technology but has to be coupled with a source of emissions, either from bionergy or from fossil fuels. Even though it is a relatively new and immature technology it quickly gained in prominence due to the possibility of reducing the carbon released from fossil power plants (Gough and Upham, 2011), allowing these to be used for a longer time even under strict temperature targets. The main risk associated with CCS is carbon leakage and thereby to the storage location. "The peer-reviewed literature that has looked at these large CCS deployment scenarios stress[es] the need for good CO2 storage site selection that would explicitly address the cumulative far-field pressure effects from multiple injection projects in a given basin." (Bruckner et al., 2014, p. 533). Further risks are mentioned in the following while evaluating BECCS. More detailed information can be found in Bachu (2008), GHG (2011) and the IPCC assessment report (Bruckner et al., 2014). Bauer et al. (2009) compare the usage of fossil CCS and BECCS and find that both would be used under an optimal climate policy but BECCS is especially desirable when trying to reach low concentration targets. In this work the combination of fossil fuels with CCS is not evaluated but is considered as an option to extend the study (see Held et al. (2009) and Edenhofer et al. (2005)).

Apart from the uncertainties regarding the individual technical potentials of bioenergy and CCS, further uncertainty stems from the type of technology used for conversion. So far "there is no single vision about where biomass is cost-effectively deployed within the energy system" (Clarke et al., 2014, p. 448). Studies that focus on this question fail to agree on the preferred type of energy (Rose et al., 2014). Comparing hydrogen and liquid fuels, Bauer et al. (2009) find that the former is more effective in removing carbon from the atmosphere. Following Humpenöder et al. (2014a), the technology named biomass to hydrogen (B2H2) is therefore used in a stylized form.

Comparing BECCS to other CDR options such as biochar and DAR, it can be considered the least cost option, as long as biomass is not restricted significantly (Kriegler et al., 2013). Nevertheless, costs represent a large degree of uncertainty depending on the level of detail to which they are modelled. Apart from production costs that are relatively cheap, aspects such as competition with food and effects on nature like soil degradation can increase costs significantly. Additionally, there have,

---

[2]Examples are saline acquifers or depleted gas fields.



so far, not been any large scale real world implementations of BECCS, that could be used as a reliable source of information to reduce the uncertainties mentioned above.

From an environmental perspective the implementation of BECCS is ambiguous. Biomass growth might induce deforestation which destroys biodiversity (Klein et al., 2014b). Being implemented responsibly with agroforestry schemes, instead, it can result in reversed effects, namely an increase in biodiversity and higher resilience to climate change (Smith et al., 2014). For a more detailed evaluation see Chum et al. (2011). Depending on the type of biomass, large-scale deployment could cause water scarcity (Klein, 2014c). Additionally, CCS storage site selection needs to be executed with caution due to the risk of carbon leakage. Leakage can negatively effect biodiversity and can have severe health impacts. Therefore, sustainability constraints should be considered.

The risks mentioned above are related to another issue when evaluating BECCS, namely a severe lack of social acceptance. On the one hand, there persists the desire on a global level to limit the "increase in the global average temperature to well below 2°C above preindustrial levels and pursuing efforts to limit the temperature increase to 1.5°C" (COP 21, 2015, p. 2). According to the literature BECCS is the most crucial technology to reach this target, as Rogelj et al. (2015) highlight that "If CDR technologies such as BECCS do not become available on a large scale and at socially acceptable costs, models are not able to limit cumulative emissions to a level that would restrict warming to 1.5°C in 2100" (Rogelj et al., 2015, p. 524). On the other hand, social acceptance is missing, especially when projects substantiate locally, a behavior titled *numbyism – not under my backyard* (The Guardian, 2009). Furthermore, the "adverse effect on food security in developing countries" through increased competition for land and thus increasing food prices needs consideration (Creutzig et al. 2012b, p. 70; citing the World Bank, 2009).

The potential of BECCS to compensate emissions across time is of particular importance in light of delayed climate policy. Models evaluating technological set ups and the impact of delayed action mostly find BECCS to be essential for feasible outcomes (Clarke et al., 2009 and Kriegler et al., 2014). Riahi et al. (2013) claim that it becomes a must for any delay after 2030. Additionally, the amount of BECCS employed increases with the length of delay (Luderer et al., 2013). The findings are independent of the type of delay: timing of political action (delayed mitigation) or divergence across countries (fragmented/delayed participation). Our work considers delay in the form of timing of political action.

## 3 Methods

### 3.1 Model

In this paper we combine the insights of two models. The main results are obtained using the integrated assessment model MIND. Additional insights are derived from the agroeconomic Global Biomass Optimization Model (GLOBIOM). They are used consecutively in a setup where GLOBIOM provides data for MIND. Similar approaches to soft linking of models can be found in Van Vuuren et al. (2013), Kriegler et al. (2013) and Fuss et al. (2013). Additional methods to link structurally similar models are described by Klein (2014c, pp. 22-23), considering the models ReMIND and MAgPIE.





MIND's economic sector follows a Ramsey-type structure which represents the economy as a whole and aims to maximize overall utility which is derived from consumption. The agricultural model is a partial equilibrium (PE) model. MIND operates in an intertemporal optimization environment. The advantage of such a combination is that more detailed information can be captured. Major influences caused by changes in the quantity of bioenergy such as land and water scarcity, biodiversity protection and prevention of soil degradation are considered in the PE model and incorporated in the price of bioenergy.

A drawback of PE models is that they by definition do not cover the whole economy, they do not depict a complete circular flow of income in the economy. Hence, cross-sector effects cannot be captured in a PE model. Instead the optimization in the PE model is performed through optimizing total surplus, i.e. consumer plus producer surplus, fulfilling the compensation principle (Chipman, 2008). Pareto efficiency and compensation principle are similar as long as no major cross-sector effects occur in the PE model. We assume that this is the case for the underlying agricultural model as the main concerns triggered by biomass growth such as land scarcity and sustainability aspects are incorporated in the model.

The Model of Investment and Technological Development (MIND) is an Integrated Assessment Model (IAM). It integrates a socio-economic system, following a Ramsey-type structure, and a climate system with the goal of evaluating the long-term impacts of climate change and the associative welfare effects of climate change policy. Additionally, it includes an explicitly modelled energy sector. This is a distinctive feature of this model, which is not present in most other IAMs. All equations are implemented as differential equations. The basic version of MIND by Edenhofer et al. (2005) has been extended by Held et al. (2009) through the incorporation of climate and technological [3] uncertainty (MIND-H). A further development (MIND-L) is conducted by Lorenz et al. (2012) who investigates learning about climate uncertainty as well as the anticipation thereof. Neubersch et al. (2014) further explores anticipatory behaviour and the resolution of uncertainty under the new framework CRA. Finally, Roth et al. (2015) extends the analysis by incorporating a delay of climate policy. Our version expands on this strand of model development by additionally including the technology option BECCS, thereby introducing a CDR technology in the MIND for the first time. Technological uncertainty (Held et al., 2009) as well as learning about climate uncertainty (Lorenz et al. (2012) and Neubersch et al. (2014)) are excluded from the analysis. The modeling language GAMS is used for the implementation of the model, using the numerical solver CONOPT.

The data on the bioenergy supply curve is generated by GLOBIOM (Schneider et al., 2011) and the Environmental Policy and Integrated Climate (EPIC) model (Izaurralde et al., 2006). The majority of the necessary information is generated using GLOBIOM running on a global scale. Results from EPIC are added to incorporate further sustainability aspects. GLOBIOM is a recursive dynamic spatial and partial equilibrium model and essentially follows the methodology of the Agricultural Sector and Mitigation of Greenhouse Gas (ASMGHG) model (Schneider et al., 2007). Its objective is to maximize total economic surplus by deriving an optimal land management strategy. It models the interplay between agricultural and forestry sectors, representing land scarcity and competition between sectors, while effects of policy changes can additionally be measured. The bioenergy price accounts for opportunity costs and land use changes, as well as different types of bioenergy and is constrained by factors such as irrigation, land and food availability. Additionally, it incorporates an additional subsidy that facilitates increased bioenergy deployment.

---

[3]Incorporated in the setup as uncertainties stemming from the cost function of renewable energy as well as size of the fossil resource base.




In this study information from the agricultural model is restricted to quantities and prices of bioenergy. The impact of bioenergy on direct emissions is negligible. We, therefore, assume carbon neutrality. Including the impact on indirect emissions would challenge the assumption of carbon neutrality. We acknowledge that the impact of bioenergy on indirect emissions, for example caused by land use change, should be considered, but here it is excluded for simplicity. This is left for a future extension of the study.

Figure 1 shows the functional fit for the whole set of data points from the agroeconomic model. It is best represented by a hyperbola of the following form $P_{\mathrm{BE}}(t) = {}^{0.9864}/[-0.0002\,Q_{\mathrm{BE}}(t)+0.1645]$ which is generated using singular value decomposition. Considering the panel on the right-hand side of Fig. 1, it is obvious that the fit for the first part of the curve is of unsatisfactory accuracy. Therefore, the procedure is repeated for the solution space of interest, a quantity of less than 600 EJ. Even though the underlying work only considers bioenergy quantities up to 325 EJ, a range up to 600 EJ is chosen deliberately to prevent a violation of the definition area of the hyperbola. The curve of the form $P_{\mathrm{BE}}(t) = {}^{-0.9713}/[0.0003\,Q_{\mathrm{BE}}(t)-0.2378]$ now adopts the shape depicted in Fig. 2. Bioenergy prices are considered to remain stable throughout time.

Using the method developed by Hoogwijk et al. (2005), van Vuuren et al. (2009) estimates prices of 2.2 and 4.8 $/GJ for amounts of 50 EJ and 125 EJ, respectively. The graphs in Fig. 2 of our study and Fig. 9 in van Vuuren et al. (2009) are similar in shape, but for our work the prices are higher. Regarding the production costs of bioenergy, GLOBIOM accounts for land-use competition which explains at least part of the mark up on prices. Klein et al. (2014b) employ a similar market setup estimating bioenergy costs of 5 $/GJ in the baseline scenario, without a tax, for a demand of 30 EJ/yr. Even though prices correspond for lower quantities, Klein et al. (2014b, Fig. 1) estimates a sharper price increase for larger bioenergy quantities. One reason is the regional resolution that allows to capture more details. Overall, the identified bioenergy supply curve lies in between the indicated literature values. Additionally, a sensitivity study on bioenergy price and quantity will shed more light on the respective impact.

### 3.1.1 Equations

The following section describes how the sectors of the model MIND have been updated to incorporate BECCS. Firstly, the budget equation in the economic sector needs the following update: investment costs of BECCS enter as a "product of specific costs and capacity additions" (Bauer et al., 2008, p. 12). The resulting budget equation in MIND is as follows: $C_t + I_{i,t} + I_{\mathrm{BECCS},t} + InNF_t \leq Y_t$[4].

Investments into BECCS ($I_{\mathrm{BECCS},t}$) are split into between the components bioenergy (BE) and CCS. It is noteworthy that bioenergy is a standalone energy option which can be used in all scenarios independent of the introduction of a climate policy. Moreover, several forms of bioenergy are considered, such as methanol, ethanol, heat, elctricity, gas and stove from wood and ethanol from corn and fatty acid methyl ester (FAME). The CCS component, in contrast cannot be deployed independently. It requires further transformation of the bioenergy. Therefore, the following separation is made: The total quantity of bioenergy is referred to as $Q_{\mathrm{BE}}$, whereas the part which is processed further is named $Q_{\mathrm{BECCS}}$.

---

[4]$C_t$− Consumption; $I_{i,t}$− Investments in sectors i; $InNF_t$− Investments traditional non fossil fuels; $Y_t$− Output





The bioenergy price $P_{BE}$ and quantity $Q_{BE}$ are determined by the supply curve (Fig. 2) and are expressed in Eq. (1). The function remains time invariant, i.e. it does not change due to yield rate increases or learning.

$$I_{BE}(t) \quad = \quad P_{BE}(t)\, Q_{BE}(t) \tag{1}$$

The bioenergy that is to be coupled to CCS requires further processing. Not every conversion method of biomass can be coupled with CCS. An overview of different conversion methods can be found in Popp et al. (2011, SI2 of the supplementary data). For simplicity a stylized conversion method is chosen in this work following Humpenöder et al. (2014a), namely biomass to hydrogen (B2H2).

$$I_{CCS}(t) = \kappa\, Q_{BECCS}(t) + \lambda\, R_{BECCS}(t) \tag{2}$$

The first part of the CCS investment costs, Eq. (2), covers the conversion process necessary to transform carbon released during the combustion process into a storable product. As MIND does not model different conversion routes from primary to secondary energy, the technology B2H2 is used in a stylized representative form (Humpenöder et al., 2014a). Associated costs amount to 9.96 \$/GJ[5] for the technology incorporating the conversion process (B2H2+CCS) and 8.25 \$/GJ for the decoupled B2H2 technology, both in current terms (Humpenöder et al., 2014a). Taking the difference, allows for an approximation of conversions costs related to CCS of 1.71 \$/GJ (=$\kappa$) which is multiplied by the quantity of bioenergy being converted ($Q_{BECCS}$). The second part covers the costs related to transport and storage of the transformed carbon ($R_{BECCS}$). Unit costs do not increase with quantity; the cost parameter is taken from Klein et al. (2014a), including costs of transportation and injection. The levelized costs amount to 9 \$/tCO2 (=$\lambda$) (Humpenöder et al. 2014a, citing Klein et al. 2014a).

Having updated the economic module of the model, the energy module is considered next. The energy equation will be updated by the addition of the positive value of bioenergy as determined by the corresponding supply curve. Including this additional technology option increases the flexibility of the model.

Finally, the emissions function of the climate module in MIND is altered. The simplified assumption regarding carbon neutrality leaves the storage process as the only component of interest. The capture rate ($\iota = 90\%$) represents the percentage of carbon that can be stored per unit of biomass ($Q_B$) burned. The data of the agroeconomic model used in this work, however, provides information on the final product, bioenergy. Therefore, an intermediary step is needed to reconvert bioenergy back into biomass Eq. (3). The conversion efficiency refers to the efficiency of transforming biomass into bioenergy. Here, we use the conversion efficiency value of B2H2 ($\omega = 55\%$), our stylized technology, for transformation. Unit conversion is accounted for by $\chi_1 = 18\,^{GJ}/_{tDM}$.

$$Q_B = \frac{Q_{BECCS}(t)}{\omega\, \chi_1} \tag{3}$$

---

[5]Humpenöder et al. (2014a) report a value of LCOE = 8 \$/GJ (B2H2+CCS) in terms of 1995 US \$. For this study the corresponding value in current dollar terms is used.



Equation 3 is incorporated into Eq. (4)[6], measuring the amount of carbon that can be removed through the usage of BECCS. Additionally, the unit of bioenergy [GJ] is converted into emission units [GtC] by $\chi_2$ ($= \mathrm{GJtoC} = 0.025\,\mathrm{tC/GJ}$)[7].

$$R_{\mathrm{BECCS}}(t) = \frac{Q_{\mathrm{BECCS}}(t)}{\omega} \, \iota \, \chi_2 \tag{4}$$

The carbon component $R_{\mathrm{BECCS}}(t)$ updates the existing emission equation $E(t)$ which is used to calculate net emissions. Gross emissions ($GE(t) = ReEx(t) + lucemi(t)$) are calculated as total emissions excluding any form of negative emissions. *ReEx* refers to the emissions that are generated in the resource extraction sector and *lucemi* is an exogenous cost vector incorporating emissions due to land use change.

$$E(t) = ReEx(t) + lucemi(t) - R_{\mathrm{BECCS}}(t) \tag{5}$$

### 3.1.2 Constraints

To represent processes such as socioeconomic or political constraints that are not explicitly incorporated in the model, gross emissions are constrained, following Lorenz et al. (2012). The scope for relative change is limited to 49% per time step. Since the model time step has a length of 5 years, this is equivalent to saying that the change in emissions cannot be larger than 13% annually (Roth, 2014, A.3.).[8]

Both components of BECCS, bioenergy as well as CCS involve uncertainties. These uncertainties are addressed by including additional constraints. For bioenergy, uncertainties mainly involve the availability of land caused by competition with food as well as whether sustainability aspects are adhered to. The IPCC assessment report (2014) provides a forecast for sustainable bioenergy potential across time (see Table 1). As a first step the respective average of these values is taken as a baseline scenario. This approach appears to be in line with the available literature. Fuss et al. (2013), for example, chose the following similar values: 41 EJ/yr (2030), 132 EJ/yr (2050) and 225 EJ/yr (2100). The constraint is implemented in a stepwise fashion, and linear functions are assumed for conjunction between the indicated points. The upper and lower bounds of the range in Table 1 are included subsequently in our sensitivity analysis.

The carbon removal potential is held at a constant level of 5.4 GtC/year, following Humpenöder et al. (2014a). The theoretical cumulative removal potential in MIND amounts to 486 GtC from 2010 until 2100, whereas the estimated usage is 388 GtC. It has to be noted that the CCS constraint cannot be binding in the short term because of the low bioenergy potential. As bioenergy deployment is independent from CCS, its full potential can be exploited despite limiting the CCS use.

The annual cap on CCS is in line with Van Vuuren et al. (2013), who estimate that maximum usage amounts to $20\,\mathrm{GtCO_2/yr}$ under their most optimistic assumption that CCS does not need a constraint on a global level. The CCS component in the study by Fuss et al. (2013) is restricted by converting the bioenergy potential through a factor of $7\,\mathrm{GJ/tCO_2}$. Such a procedure leads

---

[6]Structurally, it is similar to the *carbon removal through CCS* equation of Humpenöder et al. (2014b).

[7]This is derived from the conversion factors considered by Humpenöder et al. (2014a) who use $\chi_3 = \mathrm{tDMtoC} = 0.45\,\mathrm{tC/tDM}$ and $\chi_1 = \mathrm{tDMtoGJ} = 18\,\mathrm{GJ/tDM}$

[8]GAMS Formula: Emissions(t+1,lp,rb) - Emissions(t,lp,rb)) /(Emissions(t+1,lp,rb)+0.01) > -1,96 which is $GE(t+1) > 0,51 * GE(t)$; where GE refers to gross emissions.



to a relatively low cap of 1.6 GtC/yr in 2030, a similar cap of 5.1 GtC/yr in 2050 and a rather high cap of 8.6 GtC/yr in 2100 compared to our values. Cumulatively, the potential from 2010 until 2100 is 571 GtC, and the estimated amount used thereof totals 437 GtC (Fuss et al., 2013). Our approach is slightly more conservative, as the overall potential of CCS is lower.

### 3.1.3 Validation

Before analysing results under the comparably new decision framework CRA, the effect of BECCS using CEA[9] is shortly revised. Under CEA temporary overshooting is not allowed. Without BECCS the model cannot find a feasible solution for climate targets lower than 1.7°C, whereas with BECCS climate targets up to 1.4°C are still feasible provided immediate action is taken. Welfare losses when excluding the technology option BECCS increase by 0.81 %pts ($\Delta\%CP_{BECCS} - \Delta\%CP$), $\Delta\%$ represented in balanced growth equivalents (BGE). The difference in associative consumption losses amounts to 0.82%pts[10]

Generally, the direction of the finding is intuitive, any additional flexibility added to a model enlarges the dimensions of the solution space. In contrast, a reduction of the solution space through an additional binding constraint, such as the restriction of a technology, always leads to an increase in costs. Without BECCS, a premature switch-off of fossil fuels increases the effect of mitigation costs. With BECCS, the fossil fuel phase out is postponed, and investments are mainly placed into fossil fuels and bioenergy in the short to medium term. The prolonging effect of fossil fuels due to BECCS is the main driver for the reduced

cost impact of climate policy, when BECCS is available. Additionally, the investment share into renewables is decreased significantly, especially when considering investment peaks, leading to a smoother consumption path. A smooth consumption path across time is generally preferred because of the decreasing marginal utility of consumption. For comparison, Bauer et al. (2009) state consumption losses of 0.98 %pts.[11] One reason for identifying reduced cost savings compared to Bauer et al. (2009) can be attributed to the neglect of the transportation sector MIND. Decarbonization is generally hardest to achieve in

this sector. Luderer et al. (2012, Fig. 8 and Fig. 9) distinguish energy demand by sectors and find that BECCS is the dominant mitigation option for the transport sector, whereas solely bioenergy is used in the stationary sector. The potential gain through BECCS is therefore more valuable in ReMIND including the transportation sector.

Figure 3 displays the deployment of bioenergy or BECCS in the most basic model environment: The effect of BECCS under climate policy without delay and climate uncertainty (d) is compared to the effect of bioenergy in the BAU scenario

(b). Additionally, the respective scenarios are displayed without consideration of bioenergy (a) and BECCS (c). Already in BAU, the inclusion of bioenergy has a significant effect on total energy output, implying that bioenergy prices are already cost competitive. Under climate policy, BECCS influences the distribution of energy output even more. Whereas almost an immediate shift into renewable energy is required in the scenario excluding BECCS (c), the withdrawl from fossil fuels is postponed and even prevented in (d). Both constraint on bioenergy and that on the CCS component are binding in the long run.

Panel (d) of Fig. 3 can be compared to the conceptually similar model ReMIND (Klein, 2014c, Fig. 1.2). Their setup is different in the sense that the constraint on bioenergy is set at 300 EJ. Further constraints stem from MAgPIE, the global

---

[9]No climate uncertainty is included, instead a CS value of 3°C is employed.

[10]Based on calculated consumption until 2100, discounted at 5% (social discount rate).

[11]The graph shows for GDP and consumption the cumulative difference of a policy scenario relative to the BAU scenario discounted with 3% p.a.



land use allocation model, that provides data on bioenergy deployment for ReMIND. The models cover more detail as both are employed on a regional scale. The similarity between the results from the ReMIND model and the MIND model presented here are considered satisfactory, given that the model MIND is more simplistic. One explanation for larger bioenergy deployment in ReMIND is the less restrictive constraint on bioenergy. Lower usage of bioenergy in MIND in the long run might be due to

the fact that the underlying yield rates of bioenergy do not improve over time.

## 3.2   Decision-analytic framework CRA

The main analysis is conducted using Cost-Risk-Analysis. This is decision framework is a hybrid of CBA and CEA. The concept of climate targets is retained in a softened form and the risk of transgressing the target is weighed against mitigation costs. A risk function is used to capture the perceived risks associated with a violation of a prescribed target. The target comprises a

temperature limit, also called guard rail, of 2°C above the preindustrial level and a likelihood of reaching the climate target of 66%, this is also called the safety level. This probabilistic formulation is necessity due to the prevailing uncertainty about climate sensitivity (CS). The main advantage of CRA is the possibility of evaluating scenarios even though they might represent a transgression of the predescribed target. This is especially beneficial in light of delayed climate policy which is among the points of interest of this work. For further information regarding the decision framework CRA see Neubersch et al. (2014) and

Roth et al. (2015).

This work represents a follow-up study to Roth et al. (2015). Therefore their main results will be summarized shortly: In case of delaying climate policy by 40 years, welfare losses double when using a linear risk metric (the most conservative). The type of the risk metric plays a minor role. The risk component is the main driver of welfare losses. The utility component capturing mitigation costs even decreases with longer delays. For any delay scenario beyond 2020 the 2°C target will be transgressed at

least temporarily. The respective optimal decision for these scenarios always involves as much mitigation as possible to reduce the transgression extent. Delay is represented by no-policy (BAU) emissions during the postponement phase of climate policy (Roth et al., 2015). To ease the comparison, their results have been replicated and will be presented in the following section adjacent to the findings including the technology option BECCS.

## 4   Results

The following section deals with the main results of our analysis. The decision framework CRA is used in order to analyse the effect of BECCS in light of a delayed climate policy. The uncertainty in CS is represented through a log normal distribution of the following form: $\mathrm{pdf}(CS) = \mathcal{LN}(0.973, 0.4748)$ (Wigley and Raper, 2001) from which 20 equiprobable states of the world are sampled(Neubersch et al., 2014).



## 4.1 Temperature and Emission Paths

Figure 4 shows temperature and emission paths for different delay scenarios of climate policy, further distinguished by the deployment of BECCS. The left hand side represents the scenario excluding BECCS. The proclaimed targets of climate policy are a temperature target of 2°C and a safety level of 66%.

Considering graph (a), any delay of mitigation efforts triggers at least a minor violation of the target. A delay until 2050 even causes a global mean temperature increase of approximately 2.8°C, before the target climate policy can take effect. The reason for the transgression of the climate target can best be explained by considering respective emissions levels (c). Even though the introduction of climate risk leads to a drastic reduction of emissions, the flexibility of the emission reduction rate is limited. For a delay up to 2020 the transition to a carbon neutral economy is performed within 30 years. A faster transition is

prevented by the constraint on annual relative emission reductions of 13% implemented in MIND. This constraint represents the inertia of the energy system. This constraint is one reason why the temperature target cannot be met for a delay longer than 2025 shown in Fig. 6 (a). For the delay scenarios thereafter, the model reaches a corner solution which is caracherised by a violation of the target despite employing the maximum mitigation level.

    The effect of the negative emissions technology graph (b) is manifold: The overall temperature level in the delay scenarios

is lower, the difference between the respective temperature curves has decreased and, in addition, they jointly approach 0.5°C warming at the end of the estimation period (2200). Despite a larger mitigation portfolio, all scenarios slightly transgress the 2°C target. But scenarios with a delay until 2030 do so in a range smaller than 0.1°C. Moreover, a delay until 2050 causes global mean temperature to peak at merely 2.4°C, thus the negative impact of extreme delay scenarios can be reduced by the implementation of BECCS. This pattern is visible in all delay scenarios: The temperature peak is reduced and the temperature

target is reached with a higher probability compared to Roth et al. (2015)'s findings. Intermediary overshooting is compensated for at the end of the estimation period. By adding more flexibility to the model through including the technology option BECCS, we no longer encounter corner solutions. This can be visualized when considering marginal cost and risk functions shown in Fig. 6. The decision to transgress the target slightly is not forced through constraints but rather chosen deliberately by the optimizer due to the possibility of compensating emissions throughout time.

For scenarios including BECCS, gross (d) as well as net (e) emissions are shown to highlight the differences between bioenergy and BECCS. When comparing (c) and (d) it can be seen that gross emissions are lower when comparing (c) and (d). This is caused by the carbon neutral energy option bioenergy, whereas the difference between (d) and (e) is caused by negative emissions through BECCS which is employed for the first time in 2030.

    The peak of emissions for different delay scenarios is lower, when including the option BECCS. But the reduction of

emissions especially in the first half of the century is considerably lower. The technology option BECCS retards the fossil fuel withdrawal, resulting in higher emission levels at first which are compensated for in the second half of the century. This can also be observed when considering both temperature graphs. Roth et al. (2015) identify a slightly higher temperature increase in the first half of the century for shorter delay scenarios. The reason for this is that the start of climate policy in every delay scenario causes an immediate reduction of emissions which then causes levels of the short-lived $SO_2$ to decrease and their





cooling effect to vanish. In (b) the cooling effect remains because an immediate shut down of fossil fuels is superfluous when the possibility to compensate through CDR is enabled.

Moreover, in the scenarios using BECCS the transition to a carbon neutral economy is postponed significantly. Whereas for a delay until 2020 carbon neutrality needs to be accomplished in the year 2040 (c), BECCS postpones this shift by 60 years (d). Considering net emissions (d), a zero carbon economy is almost achieved in 2060. This is, however, achieved through compensation of emissions, not lower emissions overall. The true carbon neutrality of the economy is only achieved in 2100. Hence, BECCS can indeed be called an option to buy time (Bauer, 2005). Similarly to our findings, Van Vuuren et al. (2013) predicts net negative emissions after 2070.

Explanations can be substantiated through considering the investment decisions within the model MIND depending on the deployment of BECCS. While for the scenarios excluding BECCS investments into renewables start immediately after introducing a climate policy, the inclusion of BECCS not only defers investments into renewables, but also prevents an initial high peak. This peak is needed to compensate a substantial amount of forgone energy caused by fossil fuel withdrawal. BECCS, in turn, enables longer usage of fossil fuels because of the possibility of compensating emissions across time. There is nearly no money spent on renewables prior to 2040, if BECCS is available. The investment level into renewables decreases by around two thirds, while less than half of this amount is shifted towards deployment of BECCS, the remaining part is either freed which increases consumption (and welfare) or invested into the fossil energy production industry. The welfare effects triggered by the introduction of BECCS are analysed in the in the following section.

## 4.2 Welfare Effects

The effect of climate policy on welfare is measured by using certainty balanced growth equivalents (CBGE), which represents the "initial level of per capita consumption which, if it grows without any uncertainty at some constant rate $\alpha$ gives the same level of welfare as the expected welfare for some policy $\omega$" (Anthoff and Tol, 2009, p. 355). More specifically, in the case of uncertainty, the certainty equivalent expresses one associative amount instead of a distribution of amounts that provides the same level of expected utility (Lorenz et al., 2012). The change of CBGE, in turn, is the difference of a constant change in relative consumption between two scenarios growing at the same constant rate (Anthoff and Tol, 2009).

CBGE differences can be separated by their sources: The risk-related component is driven by an increase in temperature, whereas the utility-related part is caused by a decrease in consumption. We focus the analysis on the linear risk metric because it is the most conservative and because Roth et al. (2015) has shown that a change in the risk metric only has a minor impact. For the linear risk metric Roth et al. (2015) identify that a delay until 2050 compared to climate policy starting in 2010 causes welfare losses relative to BAU to double, from 3.55% to 7.25% (Fig. 5). This is mainly driven by the effect that the transgression of the temperature target has on the risk component. Note that the abscissa in Fig. 5 now no longer represents time but displays the different delay scenarios.

When the scenario including BECCS is considered, welfare losses decrease significantly. The main reasons are the followings. Firstly, lower investments into renewables and longer use of the cheaper fossil fuels have a positive impact on the level of consumption and thus decrease the utility-related welfare losses. Secondly, a decreased global warming lowers the risk-related





portion of losses. These effects can best be explained by examining the cumulative emissions in panel (c) and (d). Whereas for the option without BECCS the cumulative emission curves lie on top of each other in 2100 and 2200, carbon dioxide removal causes the 2200 curve to be reduced significantly. Hence, the lower the temperature, the smaller is the effect on the risk component.

To further analyse the effect of BECCS, its role in the calibration process of CRA is shortly shown. The calibration process in CRA is performed as a first step, it aligns mitigation costs and associated risk of temperature violation to an optimal emissions level that represents the climate targets ($T_\mathrm{g} = 2°C$ and $p_\mathrm{g} = 66\%$).

## 4.3    Effect of the calibration process

Figure 6 depicts the influence of delay on marginal costs (MC) and marginal risk (MR). Several MC curves are depicted
because the length of delay has an effect on costs. The longer the delay, the more mitigation is needed in a shorter period of time. The risk, in turn, is unaffected. MR is determined by using the respective cumulative emission levels, translating them into temperature paths with a linear relationship, and deriving the expected risk through the linear risk metric. The timing of emissions is not important. Roth et al. (2015) identify that for a delay beyond 2020 the model reaches a corner solution with maximum mitigation. This visualized in the intersection of MR and the MC curve 2020. The corner solution is reached
because of the fact that the MR curve lies above the MC curves, implying that the risk associated with an increase in cumulative emissions always exceeds the associated costs to avoid the risk. Considering plot (b) of Fig. 6, two things stand out. Firstly, the MR curve is significantly lower compared to the MR in graph (a). Secondly, there still exists an intersection of MR and MC curve 2050.

The cause for the first observation lies in the calibration process: The chosen normative parameters ($T_\mathrm{g} = 2°C$ and $p_\mathrm{g} =$
$66\%$) are derived from the 17th COP documented in the UNFCCC (2011). When incorporating new information into the model, such as the technology option BECCS, the conceptual question arises whether to redo calibration or not. Increasing the technology scope through BECCS influences mitigation costs. Hence, if allowed, will influence the calibration process. This decision depends upon the value system one associates with the predefined targets. Was BECCS already part of the overall picture when the targets were designed? The decision is taken in favor of modification because the importance of BECCS has
already been perceived prior to the decision-making process regarding the targets: "If negative CO2 emissions at a significant scale are not possible, then the options for meeting the limits are substantially constrained" (Den Elzen et al., 2010, p. 16). This implies that the importance of BECCS was known prior to the target definition. The second observation that an intersection exists, even for the delay 2050 scenario, means that the corner solutions can be avoided by using BECCS.

It has been demonstrated that the effect of BECCS changes patterns substantially. Several assumptions regarding the imple-
mentation of BECCS were necessary. These will be further analyzed in the next section, starting with a sensitivity study on the quantity of bioenergy and followed by evaluating the influence of the price of bioenergy.





### 4.4 Sensitivity study

Several assumptions regarding the implementation of BECCS were necessary. These will be further analyzed in the following, starting with a sensitivity study on the quantity of bioenergy and followed by evaluating the influence of the price of bioenergy.

#### 4.4.1 Quantity of bioenergy

For the sensitivity analysis, three cases are considered. The 'average BE' case refers to the amount of bioenergy used throughout this work and is the average of the range indicated by the IPCC (see Table 1). 'Min BE' and 'max BE' refer to the lower and upper bound, respectively, as indicated in the same table. Generally, the potentials in the 'average BE' and 'min BE' scenarios are fully exploited, whereas for 'max BE' it is not.

A lower bioenergy potential leads to higher temperature peaks and more transgression of the desired target. Even though
compliance with the target would theoretically be possible, the associated risks of a target transgression do not weigh heavily enough in the decision process, due to the possibility of future compensation. For 'max BE' the temperature increase can be reduced without increasing mitigation costs. The second row represents CBGE losses for the respective scenarios. The 'min BE' scenario requires alternative mitigation for the forgone bioenergy amount which slightly increases the utility component, apart from that the bioenergy potential mainly affects the losses through the risk component. Hence, the outcome is sensitive
to the assumed bioenergy potential, especially the lower potential.

#### 4.4.2 Price of bioenergy

In the first step the decision of refitting the bioenergy supply curve is tested. The first column represents the refitted bioenergy supply curve used for the main part of the analysis, whereas the second column uses the original bioenergy supply curve from Fig. 1. The third column, in turn, represents an upward shift of the refitted bioenergy supply by $10\,\$/GJ$. This number is
chosen based on other values used in the literature, such as the bioenergy prices found by Hoogwijk et al. (2005) and Klein et al. (2014b) the largest deviation of prices amounts to roughly $10\,\$/GJ$ for quantities larger than 125 EJ. This deviation is used to construct our third case.

A comparison of the price change between the original and refitted bioenergy supply curves reveals that the effect is most prominent in longer delay scenarios. The temperature overshoot in the delay 2050 scenario differs by 0.2° degrees (panel (a)
and (b)), while the associated CBGE difference amounts to 0.09%pts (panel (d) and (e)). A larger price increase as displayed in the third column has an effect on all scenarios. Whereas for lower price scenarios (column one and two) bioenergy is used immediately, a rather large bioenergy price causes its deployment to be postponed to 2050. In the short term, the economy largely relies on fossil fuels which explains the target transgression of all delay scenarios (panel (c)). This target transgression causes the risk-related CBGE loss to increase. The utility-related component is, in turn, caused by the higher price of bioenergy.
In general, although a small change of price causes some changes in the results, the investment decisions remain similar. However, a price increase of $10\,\$/GJ$ or more alters the investment decisions substantially.



## 5 Conclusions

This article investigates the effect of combining two recent innovations in integrated assessment of climate targets. (i) The decision tool 'cost risk analysis' (CRA), which is the first that allows for target transgressions and manages an evaluation of outcomes despite an underlying target transgression. (ii) Negative emission technologies, representing an important mitigation

option in case of a delay in climate policy. In stark contrast to evaluating delay within a cost effectiveness analysis (CEA) framework, where no further results can be obtained once the target is violated, CRA is ideal for analyzing the effect of delayed climate policy. In this article we represent negative emissions in terms of the technology option 'bioenergy usage in combination with carbon capture and storage' (BECCS). We combine BECCS and an evaluation using CRA and ask to what extent delay would still result in target transgression or rather target compliance once BECCS had been added to the option

portfolio.

The following conclusions can be drawn when the effects of the negative emission technology BECCS are investigated in light of delayed climate policy analysed with CRA: The inclusion of BECCS avoids corner solutions for any delay scenario, thereby extending the manoeuvring room for decision makers. BECCS has a substantially moderating effect on welfare losses. The possibility to compensate emissions across time reduces the pressure to shut down fossil energy production swiftly. As

fossil energy is the cheapest form of energy, this reduces the pressure on the utility component of welfare losses. Across delay scenarios, the cumulative emissions at the end of the simulation period (2200) are substantially lower compared to the end of this century, causing the influence of the risk component to diminish significantly. Generally, the introduction of BECCS has a smoothing effect on the investment paths. Lastly, through the sensitivity analysis we showed that the decision maker chooses to transgress no-BECCS emission scenarios and later compensate the transgression through negative emissions, depending on

the bioenergy potentials, instead of avoiding emissions in the first place. Small changes in bioenergy prices do not change investment patterns, whereas a price increase of $10\,\$/GJ$ causes investments to be placed completely differently.

How do these findings on the combination scenario 'CRA plus BECCS' compare to the individual cases to be found in the literature? For the case 'BECSS evaluated under CEA', BECCS lowers mitigation costs. However unlike the former case, even in the presence of BECCS, delay would still lead to target transgressions, however associated with risks an order of magnitude

smaller than for the no-BECCS case. Hence, numerically the CRA-based solution becomes very similar to the CEA-based solution once the mitigation cost curve has been softened by BECCS.

How does the combination scenario compare to the 'CRA, no-BECCS'-scenario? For both scenarios, and in contrast to CEA, mitigation costs decrease and risks increase with delay. However for the combination scenario, mitigation costs and climate risks are lowered by an order of magnitude. Their delay-induced changes would approximately compensate each other,

in contrast to the no-BECCS scenario characterized by a dominating risk effect. In that sense one could conclude that BECCS could 'solve' the climate problem under delay if it had no side effects and were available in an abundant way.



*Author contributions.* H.H. and J.M. jointly conceived the study, H.H. triggered this work. J.M. acquired the data and designed and implemented the simulation model, M.M.K technically supported the work, performed the consistency check of the codes, and converted MATLAB codes of bioenergy supply into GAMS codes. Interpretation was done by J.M., conceptual advice was provided by H.H. The manuscript has been written by J.M.; H.H. and M.M.K. critically revised the manuscript. M.M.K. suggested focusing on the excerpt of the bioenergy curve, the inclusion of the validation section, the sensitivity study on different prices, and the visualization of net and gross emissions.

*Competing interests.* The authors declare that they have no conflict of interest.

*Acknowledgements.* This research has, in part, been funded by the cluster of excellence "Integrated Climate System Analysis and Prediction" (CliSAP) of University Hamburg under the contract (DFG grant EXC177). Therefore, we want to express our gratitude. Additionally, we want to express our gratitude to our colleagues at FNU, with special thanks to Lukas Stein, Benjamin Blanz and Sascha Hokamp for helpful comments and fruitful discussions. Moreover, Sascha Hokamp for thorough examination of the underlying code. Furthermore, we thank Uwe Schneider and Natalie Trapp for the information on bioenergy and associative model.



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





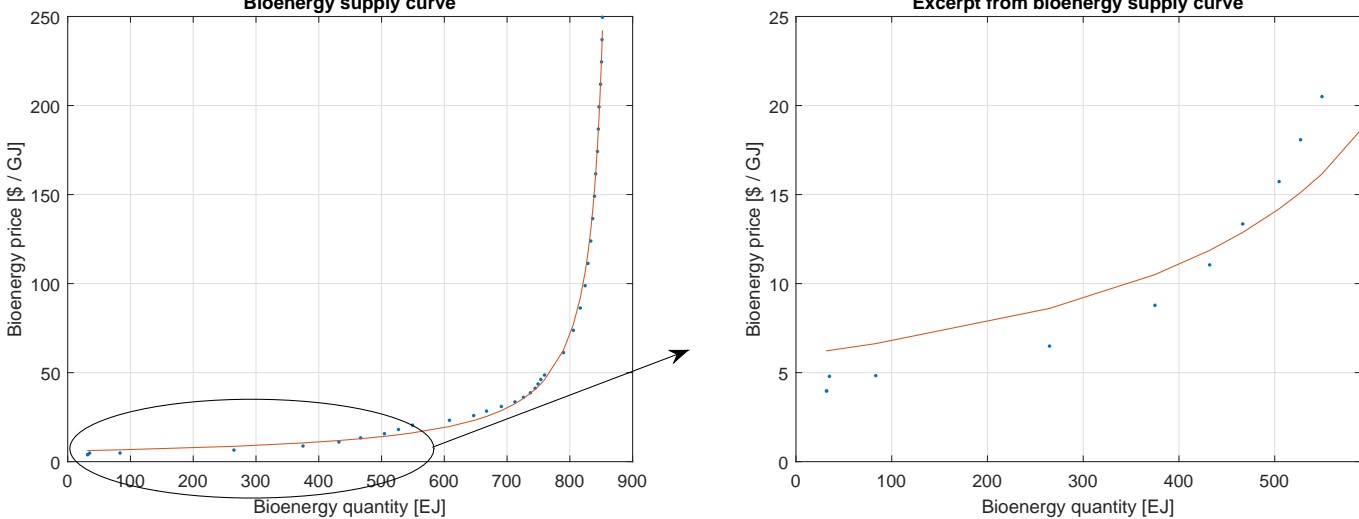

**Figure 1.** Bioenergy supply curve: Data points generated by the agroeconomic model as well as functional fit, on the left-hand side for the total set of data points, and on the ride-hand side zoomed in on the relevant quantity range.





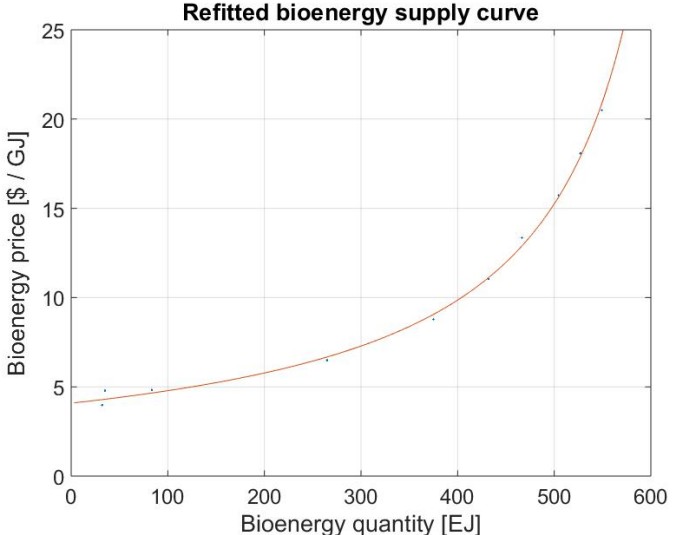

**Figure 2.** Refit of bioenergy supply curve for values < 600 EJ.





**Figure 3.** Comparing the effect of BECCS with and without climate policy displaying total amount of energy. The first two plots represents the BAU scenario, (a) without BE and (b) including BE. The plots (c) and (d) consider the 2°C target under CEA without delay, here (d) includes BECCS.





**Figure 4.** Temperature and emission paths for different delay scenarios.



**Figure 5.** Effect of delayed climate policy on welfare, distinguished by BECCS deployment ((a) without BECCS, (b) with BECCS). Changes in welfare are split by origin (red: utility-related, yellow: risk-related). Panels (c) and (d) show respective cumulative emissions in the year 2100 and 2200.





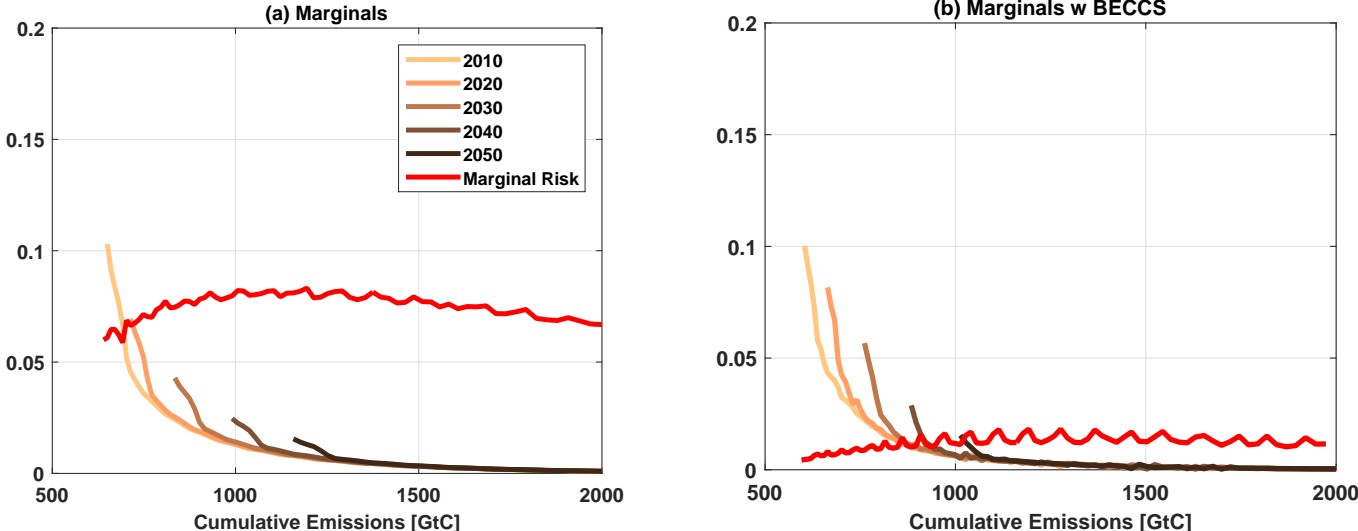

**Figure 6.** Marginal costs (MC) and marginal risk (MR) as functions of cumulative emissions, (a) without BECCS; (b) with BECCS.





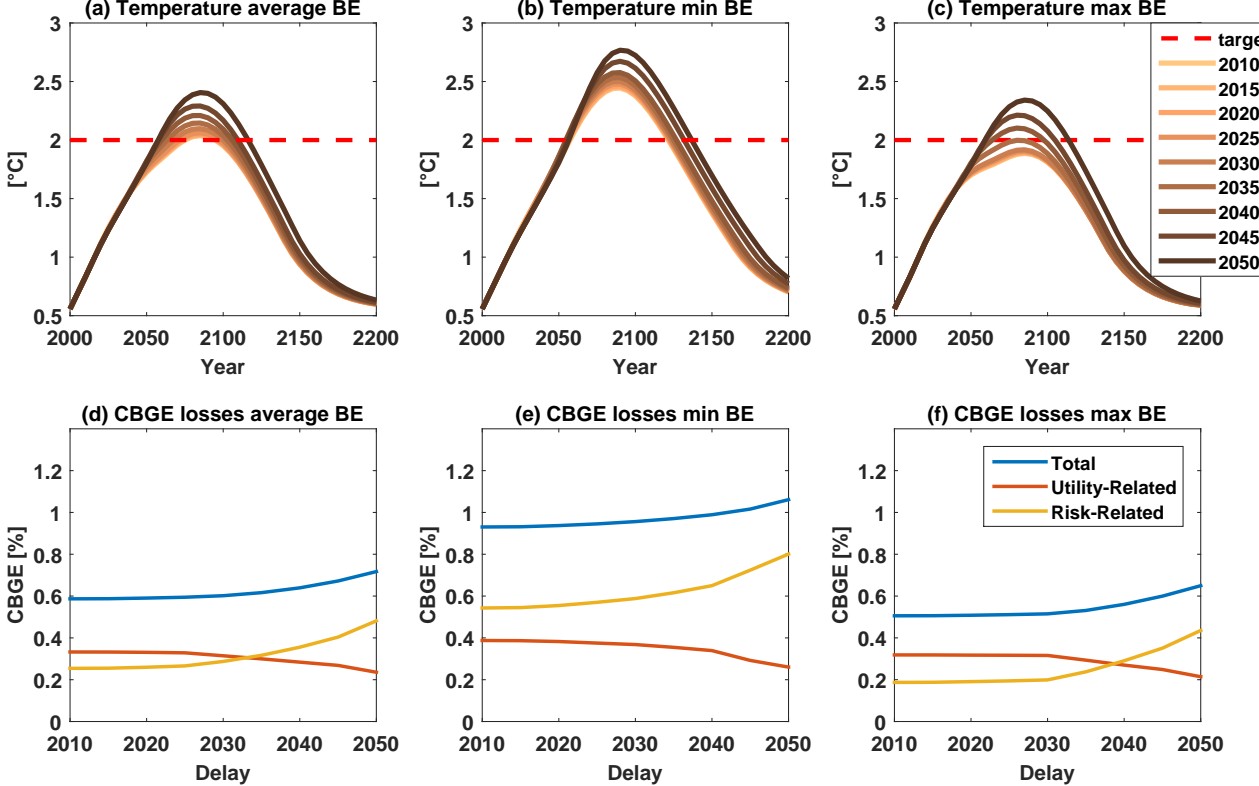

**Figure 7.** Sensitivity study on the available amount of bioenergy (BE). Average BE refers to the scenario that was considered throughout this work. Min/max BE refers to the lower/upper bound indicated in Table 1. The first row indicates different temperature paths depending on the bioenergy potential, whereas the second row shows CBGE losses in the respective scenarios (the scale is chosen differently compared to figure 5 to highlight differences between sensitivity scenarios).



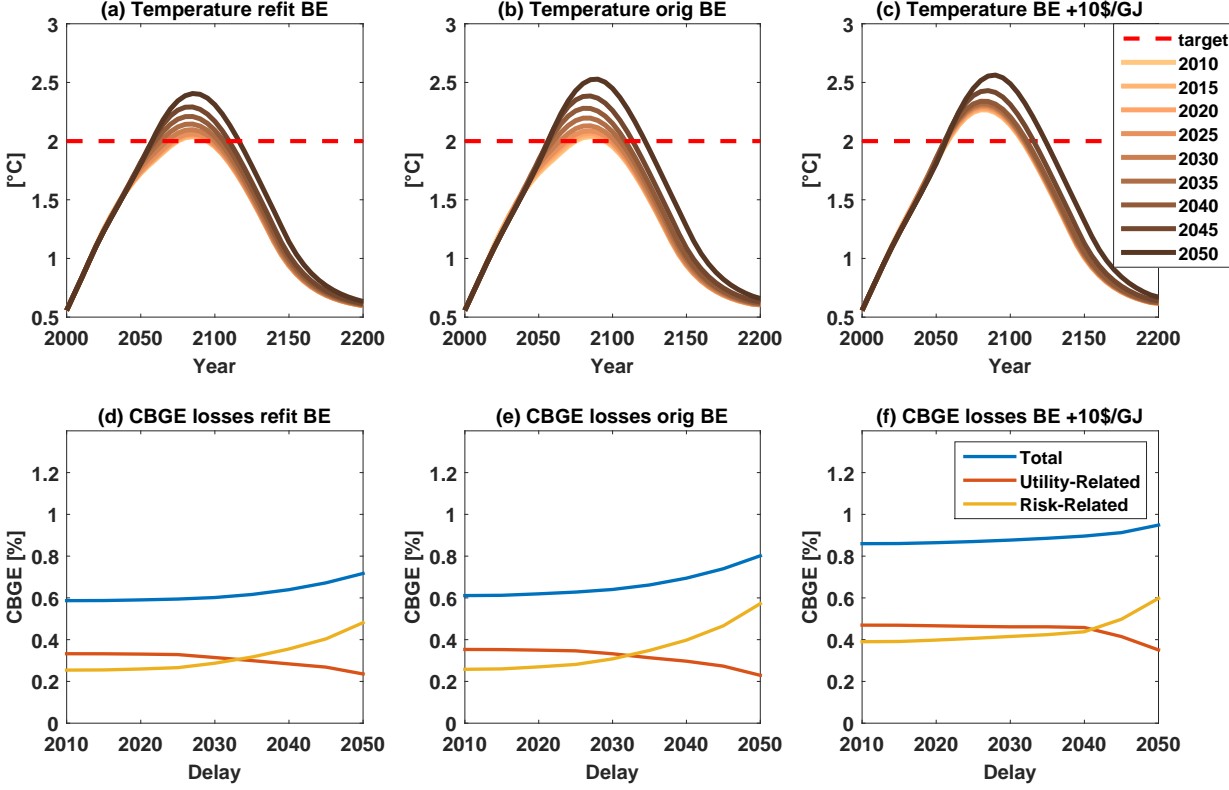

**Figure 8.** Sensitivity study on the price of bioenergy (BE). Refit BE refers to the scenario that was considered throughout this work (compare with figure 2). The second column represents the values of the original bioenergy supply curve (compare with figure 1). Column three represents a mark up of $10\,\$/\mathrm{GJ}$. The first row indicates different temperature paths depending on the respective price for bioenergy, whereas the second row shows CBGE losses in the respective scenarios (the scale is chosen differently compared to Fig. 5 to highlight differences between sensitivity scenarios).



| Year | Range | Average |
|------|-------|---------|
| 2030 | 5 - 95 | 50 |
| 2050 | 10 - 245 | 127.5 |
| 2100 | 105 - 325 | 215 |

**Table 1.** Bioenergy potential in [EJ/year] (Smith et al., 2014, p.882).