# Peer review of "The Role of Bioenergy and Carbon Capture and Storage (BECCS) in the Case of Delayed Climate Policy – Insights from Cost-Risk Analysis"

_Earth System Dynamics, 2017_

## Referee Comment (RC1) · Anonymous Referee #1 · 3 Jan 2018

General comments

This paper allows for negative emissions and BECCS with bio-energy as a representative method of CO2 removal. It uses the MIND integrated assessment model to trade off mitigation costs against the associated risks of violating temperature constraints in scenarios (tolerating a risk of 1/3 that temperature exceed 2 degrees Celsius above preindustrial). It investigates this trade-off under the assumption that there is a delay in implementation of the appropriate climate policy. This can be seen an exercise in political second best, since politicians are known to procrastinate and postpone policies. This paper does not explain why policy makers dither and postpone. It would have

been nice to use the theory of say hyperbolic discounting developed by David Laibson or a theory of political economy to get a more rigorous explanation of why policy makers postpone policies. As it is, the delay is imposed in an ad hoc manner.

Its main findings are not too surprising: BECCS avoids corner solutions that were previously identified for delayed policy scenarios and thus gives a larger window of opportunity to act, postpones mitigation efforts and thus allows longer use of fossil fuel, and curbs welfare losses by a lot. The main claim of this paper is, however, that mitigation-induced welfare losses decrease with delay whilst climate risk-induced welfare losses increase with delay by roughly the same order of magnitude. Hence, with cost-risk analysis (CRA, effectively a combination of CBA and CEA) which trades off mitigation losses against risks of overshooting temperature targets, there is a strong welfare case for BECCS in case of delayed policy implementation.

The CRA framework was first developed in Neubersch et al. (2014) and Roth et al. (2015). This study is a follow up to Roth et al. (2015), which finds that delaying climate policy by 40 years (and having business as usual with no climate policy in the run-up period) leads to a doubling of welfare losses when using a linear welfare metric, i.e., when using a zero coefficient of relative risk aversion. The metric apparently plays a minor role compared with risk itself, where uncertainty is modelled by a lognormal distribution of the climate sensitivity. For any policy implemented beyond 2020 the temperature target will be exceeded (at least) temporarily. This paper evaluates what happens when adding BECCS to the analysis of Roth et al. (2015).

Specific comments

Taking the climate science aspects as given, let us focus on the results. What we see from Figure 3, panel (c) is that emissions are on a rising business as usual path until they start to decline where the rate of decline is larger the longer the delay. Panels (d) and (e) show that, with BECCS, $CO_2$ emissions can fall more substantially. What is missing from this paper and what would be very nice to know is the required time path

of the carbon price, whether implemented via a carbon tax or a price that comes out of a competitive market for emission permits, that is needed to make sure that these emission reductions in fact take place.

If one ignores the production damages from global warming, the carbon price compatible with the safe temperature or safe carbon budget constraint should rise at a rate equal to the interest rate in view of the exhaustible nature of the carbon budget as has first been shown by William Nordhaus. In other words, the carbon price should follow a Hotelling path. However, a 2017 paper by Lemoine and Rudik in the American Economic Review allow for temperature inertia and find that the carbon price does not rise for many decades and then follows a non-monotonic pattern. In view of Ricke and Caldeira in a 2014 issue of Environmental Research Letters this latter study seems unrealistic and perhaps even irrelevant given that the time it takes for temperature to rise following a carbon impulse is only a decade. It would be good to read more on the level and shape of the time path of the carbon price that comes out of this study.

If the temperature constraint is ignored but production damages from global warming are taken account of, the carbon price rises roughly in line with GDP if damages are proportional to GDP as has been shown in a 2014 Econometrica article by Michael Golosov et al. However, if both the temperature constraint and production damages are taken account of as seems to be the case in this study, the carbon price path is a combination of these provided the temperature constraint bites. So it would be good to know what path of carbon prices delivers the immediately implemented and the various delayed emission-reduction paths shown in Figure 3. One expects the larger the delay, the higher the carbon price path needs to remain below the threshold temperature.

Another important issue the paper does not deal with is that second-best policies such as delayed policies suffer from the problem of the Green Paradox as has been forcefully pointed out by Hans Werner Sinn in his 2008 book. If fossil fuel is scarce and not abundant, fossil fuel owners anticipating a higher price of carbon in the future will pump more oil and gas today ahead of the carbon price hike. This will lead to unintended

acceleration of global warming in the short run, although admittedly more fossil fuel may be locked up in the crust of the earth. It would be good to know whether MIND has such Green Paradox effects or not. If it does, it would be good to discuss them. If it does not, it would be good to adjust MIND to allow for them where one should notice that the adverse welfare effects of such Green Paradox effects are larger if the supply of fossil fuel is less elastic and demand for fossil fuel is more elastic.

Such Green Paradox effects also lead to the issue of time inconsistency. It is important whether policy makers can commit to such a future rise in carbon prices, see a 2016 paper by Armon Rezai and Frederick van der Ploeg in Environmental and Resource Economics. Policy makers have an incentive to renege and deviate from announced plans. The challenge for future research is to investigate what second-best policies look like if policy makers cannot commit to future policies as this seems more likely and to compare these policies with those when policy makers can commit. These issues are important as these intertemporal inefficiencies might be just as important as international free rider problems. Alas, both frustrate the implementation of an ambitious climate policy.

---

## Referee Comment (RC2) · Anonymous Referee #2 · 26 Feb 2018

This article analyzes the impact of BECCS in light of delayed climate policy under CRA using the integrated assessment model MIND. The results, main insights and conclusions of the analysis does not seem to differ from usual CEA analyses and I wonder what the additional insights from using CRA really are. The conclusions the article draws are basically that i) BECCS allows more flexibility (avoids corner solutions), and ii) has a moderating effect on welfare loss because it allows a smoother transition. This seems all very well known already and could be regarded as almost trivial (you add a relatively cheap option that allows negative emissions, so what else would you expect?). The same conclusions have been made with CEA analyses many times already. Perhaps the analysis could be made more interesting if not only climate risk

is considered in the analysis, but also the risk of using BECCS itself. The latter is discussed, but not taken into account in the analysis.

Furthermore, I have some reservations regarding some of the results and assumptions of the model. Especially I do not understand why there are no renewables in the baseline up to 2080 (Fig 3a and 3b) – as currently about 20% of the global energy mix is already based on non-fossil fuels according to the IEA Energy Outlook 2017 (based on Mtoe, see p. 79). Where is the wind, solar, and hydro in the results? I also do not understand very good why first fossil fuel use decreases and then increases again in the mitigation scenarios (see Fig 3c and 3d). Finally, an important mitigation option in almost all IAMs is to increase energy efficiency, but here, there seems to be no additional improvements in energy efficiency in the mitigation scenarios. Finally, why is TNF (I guess total non-fossil, which probably means nuclear?) in mitigation scenarios the same as in baseline scenarios?

Other remarks: According to model set-up, the change in emissions is limited to 13% annually. If the restriction is indeed applied like this in the model, it would be impossible to achieve net negative emissions (if emissions are close to zero, in fact hardly any reductions are possible anymore). Why not restrict absolute reductions instead of relative reductions?

I guess I do not see a fundamental difference using CRA and using CEA with different likelihoods of achieving the target. Isn't a discussion about how much risk we are willing to take similar to the discussion on the likelihood with which we want to achieve the target? Any CEA study is under a certain assumption as to the likelihood with which a target is achieved.

In some places, more careful wording is needed: P2, line 3-5: other options than what? And I do not really agree as to whether this is an open question, as practically all 2 degree scenarios in the IPCC report assume a large share of BECCS. And the target is now well below 2 degree instead of 2C.

P3, line 2-3: Here, I don't understand the sentence, which is of critical importance as it gives the main research question. Now, it is still unclear what the precise research question actually is.

P6, line 15: I disagree here, most IAMs do model the energy sector; only the very simple cost-benefit type of IAMs do not model the energy sector explicitly. All other IAMs (GCAM, MESSAGE, WITCH, IMAGE, REMIND etc) do so to my knowedge.

P10, line 7: Here it is important to add the probability with which the target is to be achieved.

P12, line 26: "When comparing (c) and (d) it can be seen that gross emissions are lower when comparing (c) and (d)" -> not sure what you are trying to say here.

P13, line 7-8: Here a reference is made to Van Vuuren et al. (2013), stating that they predict net negative emissions after 2070. Van Vuuren et al definitely do not make predictions, but provide illustrative emissions pathways. I would also refer to more recent work, as summarized by the latest IPCC report which clearly shows the need for negative emissions as well (see Van Vuuren et al 2017 in Nature Energy Vol. 2 for a good overview on negative emissions in IPCC).

The delay scenarios seem to be very extreme: completely waiting with climate policy globally and then suddenly universal action. Why not use the Shared Policy Assumptions (SPAs) here, which imply more gradual action? This could affect results strongly, as with gradual implementation already some investment in renewables would take place.

In the beginning of the conclusions, the authors argue that one of the innovations is using negative emission technologies in integrated assessment of climate targets. I would argue this is not an innovation, since practically all 2 degree scenarios in the IPCC database incorporate such technologies already.

It is concluded that for "both scenarios, and in contrast to CEA, mitigation costs decrease and risks increase with delay". This should be formulated more carefully, I think. By definition, mitigation costs increase if you add certain restrictions and the same target has to be achieved. I think here, it is meant that not only delay is added, but also a less stringent climate target is achieved (or, in other words, a higher chance of overshoot – called here "transgression" – is allowed).

---

## Author Comment (AC1) · 30 Mar 2018

**by Anonymous Referee #1**

First of all we would like to thank the referee for carefully reviewing our manuscript. Below we will theme-wise respond to the raised issues and indicate the corresponding changes in a potential new version. We will highlight the referee's comments by italic font while our reply will be in roman font.

*General comments*

*This paper allows for negative emissions and BECCS with bio-energy as a representative method of CO2 removal. It uses the MIND integrated assessment model to trade off mitigation costs against the associated risks of violating temperature constraints in scenarios (tolerating a risk of 1/3 that temperature exceed 2 degrees Celsius above preindustrial). It investigates this trade-off under the assumption that there is a delay in implementation of the appropriate climate policy. This can be seen an exercise in political second best, since politicians are known to procrastinate and postpone policies. This paper does not explain why policy makers dither and postpone.*

We fully agree.

*It would have been nice to use the theory of say hyperbolic discounting developed by David Laibson or a theory of political economy to get a more rigorous explanation of why policy makers postpone policies. As it is, the delay is imposed in an ad hoc manner.*

This paper answers the question how society should act if the decision-analytic framework were changed from cost effectiveness analysis to cost risk analysis. Hence it contributes to answering the question how robust the results on delayed participation are as displayed in IPCC AR5 WGIII Ch6, resting on a delay-relevant subset of about 1000 scenarios, based on cost effectiveness analysis. It Roth et al., 2015, this change inverted the functional development of mitigation cost with delay, in the sense of flipping the sign of the derivative of mitigation cost with respect to delay. Here we answer the following two questions, (i) to what extent this observation is an artifact by the optimal solution being a corner solution and (ii) how the order of magnitude of cost would change if the currently most economic negative emissions technology were included. Answers to these questions we understand as our key findings. Like most papers cited in the central scenario chapter IPCC AR5 WGIII Ch6, this paper is not about explaining delay.

Will offer making this point clearer in a new ms, whereby pointing the reader to the literature the referee suggests.

*Its main findings are not too surprising: BECCS avoids corner solutions that were previously identified for delayed policy scenarios and thus gives a larger window of opportunity to act, postpones mitigation efforts and thus allows longer use of fossil fuel, and curbs welfare losses by a lot.*

We agree that these particular, cited findings are not surprising – everyone would have expected them and we would not have written a paper to convey them. Instead we would like to convey abovementioned key findings. We strive at being much clearer on what our key findings are in a revised version of the ms.

*The main claim of this paper is, however, that mitigation-induced welfare losses decrease with delay whilst climate risk-induced welfare losses increase with delay by roughly the same order of magnitude. Hence, with cost-risk analysis (CRA, effectively a combination of CBA and CEA) which trades off mitigation losses against risks of overshooting temperature targets, there is a strong welfare case for BECCS in case of delayed policy implementation.*

The former is a main claim of this paper, however, not the only one.

*The CRA framework was first developed in Neubersch et al. (2014) and Roth et al. (2015). This study is a follow up to Roth et al. (2015), which finds that delaying climate policy by 40 years (and having business as usual with no climate policy in the run-up period) leads to a doubling of welfare losses when using a linear welfare metric, i.e., when using a zero coefficient of relative risk aversion. The metric apparently plays a minor role compared with risk itself, where uncertainty is modelled by a lognormal distribution of the climate sensitivity. For any policy implemented beyond 2020 the temperature target will be exceeded (at least) temporarily. This paper evaluates what happens when adding BECCS to the analysis of Roth et al. (2015).*

We perfectly agree.

*Specific comments*

*Taking the climate science aspects as given, let us focus on the results. What we see from Figure 3, panel (c) is that emissions are on a rising business as usual path until they start to decline where the rate of decline is larger the longer the delay. Panels (d) and (e) show that, with BECCS, CO2 emissions can fall more substantially. What is missing from this paper and what would be very nice to know is the required time path of the carbon price, whether implemented via a carbon tax or a price that comes out of a competitive market for emission permits, that is needed to make sure that these emission reductions in fact take place.*

We thank the referee for this suggestion and we would implement the carbon price in a new version of the ms.

*If one ignores the production damages from global warming, the carbon price compatible with the safe temperature or safe carbon budget constraint should rise at a rate equal to the interest rate in view of the exhaustible nature of the carbon budget as has first been shown*

*by William Nordhaus. In other words, the carbon price should follow a Hotelling path. However, a 2017 paper by Lemoine and Rudik in the American Economic Review allow for temperature inertia and find that the carbon price does not rise for many decades and then follows a non-monotonic pattern. In view of Ricke and Caldeira in a 2014 issue of Environmental Research Letters this latter study seems unrealistic and perhaps even irrelevant given that the time it takes for temperature to rise following a carbon impulse is only a decade. It would be good to read more on the level and shape of the time path of the carbon price that comes out of this study.*

We perceive this a very exiting discussion and are happy to elaborate on it in a new version of our ms in view of our numerical data.

*If the temperature constraint is ignored but production damages from global warming are taken account of, the carbon price rises roughly in line with GDP if damages are proportional to GDP as has been shown in a 2014 Econometrica article by Michael Golosov et al. However, if both the temperature constraint and production damages are taken account of as seems to be the case in this study, the carbon price path is a combination of these provided the temperature constraint bites. So it would be good to know what path of carbon prices delivers the immediately implemented and the various delayed emission-reduction paths shown in Figure 3. One expects the larger the delay, the higher the carbon price path needs to remain below the threshold temperature.*

We agree with the latter statement in case cost effectiveness analysis was utilized. However cost risk analysis employs a somewhat 'permeable' upper limit in temperature, somewhat resembling a second order phase transition in thermodynamics. This induces a decline in mitigation cost. As the latter is a convolute of carbon price, discounting, and further effects, it is subject to further investigation how, in fact, the carbon price would evolve under delay. We offer delivering such a discussion in a new version of the ms.

*Another important issue the paper does not deal with is that second-best policies such as delayed policies suffer from the problem of the Green Paradox as has been forcefully pointed out by Hans Werner Sinn in his 2008 book. If fossil fuel is scarce and not abundant, fossil fuel owners anticipating a higher price of carbon in the future will pump more oil and gas today ahead of the carbon price hike. This will lead to unintended acceleration of global warming in the short run, although admittedly more fossil fuel may be locked up in the crust of the earth. It would be good to know whether MIND has such Green Paradox effects or not. If it does, it would be good to discuss them. If it does not, it would be good to adjust MIND to allow for them where one should notice that the adverse welfare effects of such Green Paradox effects are larger if the supply of fossil fuel is less elastic and demand for fossil fuel is more elastic.*

The Green Paradox is primarily an effect induced by de-central dynamics (resource owners versus regulators). However even in a social planner model like the MIND model, part of this effect might show up as there are different versions of how to model 'delay'. We can distinguish a policy-anticipating from a non-anticipating version. In the anticipating version in turn society might use more fossil fuel than for the baseline scenario, as it anticipates scarcity of fossil resources being relaxed after activation of a climate policy. All of this discussion is ignored in the current version of the ms and shall be taken on board in a revised version of the ms. Furthermore, we offer also adding a scenario of above-social

planner-level usage of fossil fuels prior policy, to somewhat mirror the case of an unmitigated Green Paradox effect.

*Such Green Paradox effects also lead to the issue of time inconsistency. It is important whether policy makers can commit to such a future rise in carbon prices, see a 2016 paper by Armon Rezai and Frederick van der Ploeg in Environmental and Resource Economics. Policy makers have an incentive to renege and deviate from announced plans. The challenge for future research is to investigate what second-best policies look like if policy makers cannot commit to future policies as this seems more likely and to compare these policies with those when policy makers can commit. These issues are important as these intertemporal inefficiencies might be just as important as international free rider problems. Alas, both frustrate the implementation of an ambitious climate policy.*

We agree that these are important issues. However our paper is from a social planner's perspective, thereby responding to IPCC AR5 WGIII Ch6. Our ambition is not to explain delay, but to recommend action, given delay, for a new, dynamically consistent decision-analytic framework (i.e. cost risk analysis). We would make this clearer in a new version of our ms and cite the abovementioned literature.

---

## Author Comment (AC2) · 30 Mar 2018

**by Anonymous Referee #2**

First of all we would like to thank the referee for critically reviewing our manuscript. Below we will theme-wise respond to the raised issues and indicate the corresponding changes in a potential new version. We will highlight the referee's comments by italic font while our reply will be in roman font.

*This article analyzes the impact of BECCS in light of delayed climate policy under CRA using the integrated assessment model MIND. The results, main insights and conclusions of the analysis does not seem to differ from usual CEA analyses and I wonder what the additional insights from using CRA really are. The conclusions the article draws are basically that i) BECCS allows more flexibility (avoids corner solutions), and ii) has a moderating effect on welfare loss because it allows a smoother transition. This seems all very well known already and could be regarded as almost trivial (you add a relatively cheap option that allows negative emissions, so what else would you expect?). The same conclusions have been made with CEA analyses many times already. Perhaps the analysis could be made more interesting if not only climate risk is considered in the analysis, but also the risk of using BECCS itself. The latter is discussed, but not taken into account in the analysis.*

The conclusions the referee draws as main conclusions from our paper are indeed not surprising and we would not have written a paper to convey them. However this paper mainly strives at another discussion: how society should act if the decision-analytic framework were changed from cost effectiveness analysis to cost risk analysis. It Roth et al., 2015, this change inverted the functional development of mitigation cost with delay, in the sense of flipping the sign of the derivative of mitigation cost with respect to delay. Here we answer the following two specific questions, (i) to what extent this observation is an artifact by the optimal solution being a corner solution and (ii) how the order of magnitude of cost would change if the currently most economic negative emissions technology were included. Answers to these questions we understand as our key findings. Hence part of the phenomena discussed in our ms is in stark contrast to what we know from cost effectiveness analysis. A revised version of our ms would make much clearer what our key findings are. However taking BECCS risk into account would lead into a very different discussion making a like-with-like intercomparison with Roth et al. difficult.

*Furthermore, I have some reservations regarding some of the results and assumptions of the model. Especially I do not understand why there are no renewables in the baseline up to 2080 (Fig 3a and 3b) – as currently about 20% of the global energy mix is already based on*

*non-fossil fuels according to the IEA Energy Outlook 2017 (based on Mtoe, see p. 79). Where is the wind, solar, and hydro in the results? I also do not understand very good why first fossil fuel use decreases and then increases again in the mitigation scenarios (see Fig 3c and 3d). Finally, an important mitigation option in almost all IAMs is to increase energy efficiency, but here, there seems to be no additional improvements in energy efficiency in the mitigation scenarios. Finally, why is TNF (I guess total non-fossil, which probably means nuclear?) in mitigation scenarios the same as in baseline scenarios?*

The baseline scenario has not been updated to include an elevated level of renewables to preserve comparability with older studies of the model. A new version of our ms would offer a respective adjustment as a sensitivity study. The MIND model is a stylized model to include endogenous technological change, hereby being more realistic on mitigation cost than e.g. Nordhaus' DICE model. It is the optimal model to study stylized effects when decision-making uncertainty is to be included (in our case on climate system response to greenhouse gas-induced forcing of the climate system). The MIND model by definition does not resolve the renewable sector any further. It is the underlying technical innovation of this paper to explicitly model BECCS within MIND in order to represent the option of negative emissions. Regarding the tradition of all MIND-based papers we prefer sticking to leaving the exogenous TNF (in fact nuclear energy, large hydropower, and traditional biomass policy) untouched to compare like with like. We would make much clearer in a revised version of the ms that these days, the MIND model is not meant to compete with high-resolution energy system, yet also intertemporally optimizing integrated assessment models like REMIND or WITCH, but to demonstrate order-of-magnitude effects when changing the analytic framework for decision-making under uncertainty.

*Other remarks: According to model set-up, the change in emissions is limited to 13% annually. If the restriction is indeed applied like this in the model, it would be impossible to achieve net negative emissions (if emissions are close to zero, in fact hardly any reductions are possible anymore). Why not restrict absolute reductions instead of relative reductions?*

This relative restriction origins from Kriegler & Bruckner, Climatic Change, 2004, who found a relative restriction more intuitive than an absolute restriction. The rationale behind this is a convex 'social and infrastructure change-induced cost' curve, in analogy to a convex mitigation cost curve. The referee is perfectly right in that if such a restriction were applied to the net emissions, no net negative emissions were possible. However we do apply it to the gross emissions for which the restriction was invented. We would make this subtle, yet crucial point much clearer in a new version of the ms.

*I guess I do not see a fundamental difference using CRA and using CEA with different likelihoods of achieving the target. Isn't a discussion about how much risk we are willing to take similar to the discussion on the likelihood with which we want to achieve the target? Any CEA study is under a certain assumption as to the likelihood with which a target is achieved.*

We agree with the referee that these days – at the latest since the Copenhagen Diagnosis as of 2009 – CEA is always meant to be seen as an approximation of CCP (chance constraint programming, i.e. optimization under a probabilistic climate target, as no temperature target can be complied with for 100%). However the key point of CRA is not usage of a probabilistic target, but the fact that this probabilistic target, for the first time, is

made tradable against mitigation cost, while in CEA/CCP it remains a rigid target. In the language of decision theory: the change from CEA/CCP to CRA is the change from a lexicographic (climate first) preference order to a non-lexicographic, utilitarian one. A rigid target is replaced by a somewhat 'porous' one – it somewhat resembles a second order phase transition from thermodynamics. This is the core reason why delay has just the opposite effect in CEA as against CRA. We would make this central point much clearer in a new version.

…

Regarding the more detailed comments made by the referee we will comply with all the suggestions – except for the last comment which re-iterates above misunderstanding of CRA: we in fact have to insist that mitigation cost decline, the reason being that we have replaced a strict by a soft target, the very point of this and the Roth et al. paper.

Overall we are convinced that we can address most of the referee's concerns in a new version. However we ask for the referee's understanding that we cannot have a detailed energy system discussion here as it would be possible for high-resolution models like REMIND or WITCH. Here it is about to demonstrating the key effect of changing the decision-analytic framework addressing uncertainty, in the very presence of BECCS.